# Contribution of *ADD3* and the HLA Genes to Biliary Atresia Risk in Chinese

**DOI:** 10.3390/ijms241914719

**Published:** 2023-09-29

**Authors:** Meng-Meng Cui, Yi-Ming Gong, Wei-Hua Pan, Hao-Yue Pei, Mei-Rong Bai, Huan-Lei Song, Xin-Ru Han, Wen-Jie Wu, Wen-Wen Yu, Bei-Lin Gu, Wei Cai, Ying Zhou, Xun Chu

**Affiliations:** 1Department of Pediatric Surgery, Xinhua Hospital Affiliated to Shanghai Jiaotong University School of Medicine, Shanghai 200092, China; cuimengmeng510@163.com (M.-M.C.); mecerd@163.com (Y.-M.G.); panweihua@xinhuamed.com.cn (W.-H.P.); wuwenjie@xinhuamed.com.cn (W.-J.W.); caiw1978@163.com (W.C.); 2Shanghai Institute of Pediatric Research, Shanghai 200092, China; peihaoyue1204@163.com (H.-Y.P.); 15221095129@163.com (M.-R.B.); songhuanlei@xinhuamed.com.cn (H.-L.S.); hanxr7@163.com (X.-R.H.); yww_www@163.com (W.-W.Y.); gpl207@126.com (B.-L.G.); 3Shanghai Key Laboratory of Pediatric Gastroenterology and Nutrition, Shanghai 200092, China

**Keywords:** biliary atresia, genome-wide association study, cytomegalovirus, *ADD3*, HLA

## Abstract

Nonsyndromic biliary atresia (BA) is a rare polygenic disease, with autoimmunity, virus infection and inflammation thought to play roles in its pathogenesis. We conducted a genome-wide association study in 336 nonsyndromic BA infants and 8900 controls. Our results validated the association of rs17095355 in *ADD3* with BA risk (odds ratio (OR) = 1.70, 95% confidence interval (95% CI) = 1.49–1.99; *p* = 4.07 × 10^−11^). An eQTL analysis revealed that the risk allele of rs17095355 was associated with increased expression of *ADD3*. Single-cell RNA-sequencing data and immunofluorescence analysis revealed that *ADD3* was moderately expressed in cholangiocytes and weakly expressed in hepatocytes. Immuno-fluorescent staining showed abnormal deposition of ADD3 in the cytoplasm of BA hepatocytes. No ADD3 auto-antibody was observed in the plasma of BA infants. In the HLA gene region, no variants achieved genome-wide significance. HLA-DQB1 residue Ala57 is the most significant residue in the MHC region (OR = 1.44, 95% CI = 1.20–1.74; *p* = 1.23 × 10^−4^), and HLA-DQB1 was aberrantly expressed in the bile duct cells. GWAS stratified by cytomegalovirus (CMV) IgM status in 87 CMV IgM (+) BA cases versus 141 CMV IgM (−) BA cases did not yield genome-wide significant associations. These findings support the notion that common variants of *ADD3* account for BA risk. The HLA genes might have a minimal role in the genetic predisposition of BA due to the weak association signal. CMV IgM (+) BA patients might not have different genetic risk factor profiles compared to CMV IgM (−) subtype.

## 1. Introduction

Biliary atresia (BA) is a fibrosing obstructive cholangiopathy affecting both the extrahepatic and intrahepatic bile ducts that present exclusively in newborns and early infants [1]. Although children appear normal at birth, they rapidly progress to cholestasis, hepatic fibrosis and eventually liver failure in the first several months of life [2]. BA is a rare disease with a prevalence of 0.5 to 0.8 per 10,000 live births in western countries [2,3]. Whereas the incidence is considerably higher in Asia population, with 1.1 per 10,000 births in Japan [4] and 1.5 per 10,000 live births in Taiwan [5].

BA exists in two proposed types: the acquired/perinatal form and the embryonic/congenital form [6]. The rare embryonic form (~15% of Caucasian cases) is associated with other congenital anomalies, and may be due to defective development of the extrahepatic biliary tract caused by genetic or somatic mutations [1]. The isolated BA represents the vast majority of BA cases (~85% of cases), which is also referred to as nonsyndromic BA. It is believed to be a multifactorial disorder, in which environmental agents, such as toxins and viral infection, interact with genetic factors that influence susceptibility [6,7,8,9]. A prevailing hypothesis for the pathogenesis of BA suggests that an initial virus infection insults the bile duct epithelia, triggering a secondary autoimmune-mediated injury that leads to progressive bile duct injury and cirrhosis [2,6]. Multiple hepatotropic RNA viruses were implicated in the etiology of BA, with cytomegalovirus (CMV) often considered as a trigger, albeit inconclusively. CMV IgM (+) BA was identified as a distinct clinical entity, presenting a poorer prognosis compared to CMV IgM (−) patients [8].

Recent genome-wide association studies (GWASs) have uncovered susceptibility genes for nonsyndromic BA. The first GWAS for BA in Chinese population identified the top associated SNP rs17095355 located in the intergenic region between Adducin 3 (*ADD3*) and X-prolyl aminopeptidase 1 (*XPNPEP1*); subsequent fine-mapping and expression analysis revealed ADD3 as the susceptibility gene [10,11]. These associations were replicated in Thai, Chinese and Caucasians [12,13,14,15]. *Add3* dysfunction in zebrafish caused phenotypes resembling BA [16]. A GWAS involving 61 patients and 5088 healthy controls from mixed population uncovered deletions in 2q37.3 containing *GPC1* in BA patients [17,18]. Common SNPs in GPC1 demonstrated modest associations with BA in Chinese case-control sample sets [15,19]. Common SNPs in the 3′ flanking region of *ARF6* were associated with BA risk in a GWAS with 80 Caucasian BA cases and 2818 controls [20]. A GWAS in a European–American population established an association between EFEMP1 and BA risk in 343 nonsyndromic BA patients and 1716 controls; this finding was validated in an independent European–American cohort comprising 156 patients with BA and other extrahepatic anomalies and 212 controls [21].

Human leukocyte antigen (HLA) is the strongest genetic factor associated with autoimmune and inflammatory diseases. However, reported associations between HLA alleles and BA susceptibility have been conflicting [22,23,24,25]. A Brazilian study in 55 BA patients with or without major extrahepatic congenital anomalies revealed that the frequency of HLA-B12 is significantly higher in patients without associated malformations [22]. No associations between HLA alleles with BA susceptibility were found in a study of 101 Northern European BA children compared with 137 controls [23]. However, HLA-A24-B52-DR2 haplotype showed an association with BA in a Japanese study with 392 BA patients [24]. In a more recent study of 180 BA patients including African, Caucasian, Hispanic and Asian populations and 360 racially matched controls, no HLA allele or shared epitope association was identified in BA [25]. HLA association with BA was not reported within the GWAS analysis [10,17,20,21].

We have previously conducted a replication study for common variation of *ADD3*, *GPC1*, *ARF6* and *EFEMP1*, and validated associations of *ADD3* and *GPC1* SNPs with BA risk in Chinese population [15]. In the current study, we performed a GWAS in a Chinese sample set to explore common variation contributing to BA risk, and fine mapped the HLA region by the imputation approach in our GWAS data. Our results confirmed the contribution of common variants to BA susceptibility. However, our results did not support that the common variants in the HLA region made a substantial contribution to the susceptibility of BA risk. To dissect the molecular mechanisms of susceptibility genes involved in the pathogenesis of BA, we integrated the GWAS results with previously published single-cell RNA sequencing data of the liver, publicly available expression quantitative trait locus (eQTL) data and experimental analysis for a better comprehension of gene functions. Furthermore, we employed stratification analysis to clarify whether different underlying genetic variation existed between CMV IgM (+) and CMV IgM (−) BA patients.

## 2. Results

### 2.1. Summary of the GWAS Results

A total of 336 nonsyndromic BA cases and 8900 of Han Chinese were genotyped (Appendix A). After quality control, we tested 398,532 single genetic variants for association with BA susceptibility in 317 sporadic BA cases and 8843 controls of Han Chinese (Appendix A). To minimize the adverse impact of population stratification, association analysis was performed using the top five PCs as covariates via logistic regression. Two SNPs in the *ADD3* gene region at 10q25 showed conventional genome-wide significant association with BA (*p* < 5.0 × 10^−8^; Figure 1A, Table 1). No SNPs in the HLA gene region reached the genome-wide significance.

### 2.2. Association of SNPs in ADD3 Gene Region with BA Susceptibility

The strongest association signal was detected in the *ADD3* gene region at chromosome 10q25 (Figure 1A and Figure 2A). Two SNPs in strong linkage disequilibrium (r^2^ = 0.71) surpassed the genome-wide significant threshold, namely rs17095355 about 20 kb 5′ upstream *ADD3* (odds ratio (OR) = 1.70, 95% confidence interval (95% CI) = 1.49–1.99; *P*_Logistic_ = 4.07 × 10^−11^) and rs10884919 in an intronic region of *ADD3* (OR = 1.60, 95% CI = 1.36–1.87; *P*_Logistic_ = 5.51 × 10^−9^) (Table 1 and Appendix A). Consistent with previous GWAS, rs17095355 was the most associated SNPs with BA [10]. The association of rs10884919 was not significant when conditioning on rs17095355 (*p* = 0.62), whereas rs17095355 was significantly associated with BA when conditioning on rs10884919 (*p* = 1.93 × 10^−3^), suggesting a single association signal in this region. Imputation analysis revealed that additional 153 SNPs in this region showed genome-wide association with BA (*p* < 5.0 × 10^−8^; Appendix A).

### 2.3. Risk Allele of rs17095355 Is Associated with Increased Expression of ADD3

Rs17095355 and these correlated SNPs fall within a region with strong enhancer activity and DNase I hypersensitivity sites annotated by Haploreg, suggesting a potential regulatory role for these SNPs. We performed eQTL analyses for these associated SNPs using data from the liver, whole blood and spleen tissues obtained from GTEx, which might be relevant tissues for BA pathogenesis. The risk allele T of rs17095355 was associated with increased expression of *ADD3* in the spleen (*P*_GTEx_ = 5.1 × 10^−13^) and whole blood (*P*_GTEx_ = 8.2 × 10^−9^), but not in the liver (Figure 2B,C, Appendix A). However, an associated SNP, rs3862007, was the strongest eQTL for *ADD3* in the liver (*P*_GTEx_ = 3.6 × 10^−7^), spleen (*P*_GTEx_ = 9.1 × 10^−20^) and whole blood (*P*_GTEx_ = 1.2 × 10^−13^), and the risk allele G was associated with increased expression of *ADD3*. Rs3862007 was in moderate LD with rs17095355 (r^2^ = 0.66) and showed association with BA from imputation analysis (*P*_Additive_ = 4.71 × 10^−9^). Of note, the risk alleles of both rs17095355 and rs3862007 were associated with increased expression of *ADD3* in most of the human body tissues (Appendix A). There was no evidence against colocalization of the eQTL and BA association, thus supporting *ADD3* as a candidate causal gene. Together, these indicated that the upregulation of *ADD3* increased the risk of BA.

### 2.4. ADD3 Aberrantly Deposits in the Cytoplasm of BA Livers

*ADD3* is ubiquitously expressed in adult human tissues [26].

In silico expression analysis using the GTEx database showed a relatively low expression level of *ADD3* in liver tissues (Appendix A). Our immunofluorescence (IF) results confirmed *ADD3* expression in cholangiocytes and hepatocytes, consistent with previous studies (Figure 3A and Appendix A) [11,27]. We further compared the appearance of ADD3 in control and BA liver sections using a immunofluorescence microscopy. In both control and BA liver, ADD3 outlined the periphery of the cell membranes in a perimembranous staining pattern. However, ADD3 outlined the periphery of the cell membranes in a darker and wider pattern in the intrahepatic bile ducts of controls compared with the tortuous and distorted proliferating bile ducts of BA patients (Figure 3A–C). Interestingly, dot-like ADD3 deposits were observed in the cytoplasm of BA hepatocytes, but no ADD3 deposits were observed in the cytoplasm of BA cholangiocytes (Figure 3A–C). ADD3 is a peripheral membrane protein that caps actin filaments. The cholangiocyte cytokeratin 7 (CK7) is an intermediate filament of cholangiocytes. The localization of two major cytoskeletal elements, actin and cytokeratin, is most notable in the subplasmalemmal regions. We thus examined the colocalization of ADD3 and CK7 in cholangiocytes. A lower degree of colocalization of ADD3 and CK7 in BA cholangiocytes compared with CC was observed, it is tempting to speculate that the location of ADD3 changed in BA cholangiocytes (Figure 3B,C). In addition, we used publicly available single-cell RNA sequencing data to examine the cell specific expression of *ADD3* in liver tissue, and found that *ADD3* was moderately expressed in cholangiocytes and weakly expressed in hepatocytes (Figure 3D).

### 2.5. Assay of Anti-ADD3 Antibodies

It was proposed that an aberrant autoimmune response targeting cholangiocytes was involved in the etiology of biliary atresia. Autoantibodies in human BA were detected in previous studies [28,29]. Since *ADD3* showed strong association with BA susceptibility, we asked whether autoantibody against ADD3 existed in BA patients. Thus, we performed ELISA assays to measure the levels of auto-antibodies against the ADD3 antigen in the plasma of BA patients and controls. We did not find significant differences among the three groups according to the OD values (*p* > 0.05) (Figure 3E). These results suggested that the attribution of aberrant expression and distribution of ADD3 to BA etiology might not be caused by autoimmune response.

### 2.6. Summary of Imputation Results for the HLA Gene Region

We carried out imputation analysis for *HLA* genes using 6530 GWAS SNPs from ~29 Mb to ~35 Mb on chromosome 6p21.3 in 317 BA cases and 8843 controls. The Pan-Asian reference panel was used as reference in the imputation analysis with SNP2HLA software (v1.0.3) [30,31,32]. A total of 5321 markers with minor allele frequency ≥1% and Plink INFO ≥ 0.8 were produced from the imputation, including 4256 SNPs, 172 2- and 4-digit classical alleles, and 893 amino acid residues for the classical *HLA* genes (Figure 4A, Appendix A).

### 2.7. Associations of Classical HLA Genes with BA

None of the variants resulted from the imputation in the *HLA* region surpassed the genome-wide threshold (Figure 4A). We found no evidence of epistatic interactions between risk variants of *ADD3* and any of the *HLA* alleles. The top signal was a cluster of SNPs in high LD falling in the region between the HLA I and II gene clusters, with the best SNP rs707936 (OR = 1.69, 95% CI = 1.35–2.11; *P*_Logistic_ = 3.46 × 10^−6^) about 400 kb upstream of HLA-B (Table 1).

HLA-DQB1 residue Ala57 was the most significant HLA residue (OR = 1.44, 95% CI = 1.20–1.74; *P*_Logistic_ = 1.23 × 10^−4^) and residue Asp57 showed a protective effect (OR = 0.75, 95% CI = 0.63–0.88; *P*_Logistic_ = 4.32 × 10^−4^) (Table 2, Appendix A). Nine HLA-DQB1 residues (Ser28-Ser30-Ile37-Glu46-Phe47-Leu52-Leu55-Lys71-Ala74) in completely perfect LD showed similar significance (OR = 1.46, 95% CI = 1.18–1.81; *P*_Logistic_ = 4.58 × 10^−4^), which was encoded by HLA-DQB1*0201 (Figure 3B, Table 2). HLA-DQB1*0201 was the most significant 4-digit HLA allele (OR = 1.46, 95% CI = 1.18–1.81; *P*_Logistic_ = 4.75 × 10^−4^). HLA-DR2 haplotype showed an association with BA in the Japanese population [24]. In the current Chinese population, HLA-DRB1*15 was associated with protection of BA with a nominal significance (OR = 0.65, 95% CI = 0.50–0.84; *P*_Logistic_ = 9.41 × 10^−4^), which were tagged by HLA-DRB1 Ala71 and Ser1 (OR = 0.65, 95% CI = 0.50–0.84).

The most significant amino acid residue of *HLA*-I gene was HLA-B Leu103 (OR = 1.34, 95% CI = 1.13–1.58; *P*_Logistic_ = 6.76 × 10^−4^) and HLA-B*58:01 was the top classical *HLA-B* allele (OR = 1.58, 95% CI = 1.19–2.10; *P*_Logistic_ = 1.62 × 10^−3^) (Figure 4A,B, Table 2). HLA-B12 and B52 were reported association with BA in previous studies [22,24].

HLA class I genes are expressed in normal bile duct epithelium. However, strong expression of class II HLA-DR was found on the liver bile duct epithelia of BA patients [33,34]. Since HLA-DQ variants showed the strongest association among the HLA region in the current study, we further investigated whether HLA-DQ antigens were aberrantly present on bile duct epithelia of BA patients. Our results found HLA-DQ expression and confirmed HLA-DR expression in the intrahepatic biliary epithelium in BA, whereas no expression was detected in controls, inconsistent with previous reports (Figure 4C,D) [33,34].

### 2.8. Stratification Analysis by CMV IgM

CMV IgM (+) BA has been proposed as a distinct clinical and pathological entity with a poorer outcome compared with CMV IgM (−) BA [8]. In the current sample set, positive CMV serology was found in 38.2% (87/228) of the BA infants. We investigated whether CMV IgM (+) BA and CMV IgM (−) BA had different genetic structures using GWAS data. No SNPs surpassed the genome-wide association significance threshold in GWAS of 87 CMV IgM (+) BA cases or 141 CMV IgM (−) BA cases comparing with 8843 controls (Figure 1B,C). We also performed GWAS for 87 CMV IgM (+) BA cases versus 141 CMV IgM (−) BA cases, and found no variants achieved genome-wide association significance (Figure 1D). The most associated BA risk SNP rs17095355 in the *ADD3* gene region had no distribution difference between the IgM (+) BA and CMV IgM (−) BA groups (*p* > 0.05).

## 3. Discussion

Using a GWAS study and subsequent imputation analysis for BA risk in the Chinese population, we confirmed the association of *ADD3* with genome-wide significance and found associations of HLA-DQB1 residue Ala57 and HLA-B Leu103 association with a nominal significance. The eQTL analysis revealed that risk alleles of susceptibility SNPs correlated with increased expression of *ADD3*. Integrating available scRNA data and IF results, we confirmed the expression of *ADD3* in cholangiocytes and hepatocytes. Aberrant cytoplasmic deposition of ADD3 was found in hepatocytes of BA lives. Aberrant expression of HLA-DQ and DR was observed in the intrahepatic bile duct epithelium of BA patients.

Our previous study with 340 BA patients and 1665 controls investigated the four reported GWAS loci including *ADD3*, *GPC1*, *ARF6* and *EFEMP1*, which validated the association of *ADD3* variants with study-wide significance and the association of *GPC1* variants with nominal significance [15]. The current GWAS analysis used a case cohort partially overlapped with the previous case cohort, and a relatively larger control cohort to increase statistical power. The *ADD3* SNPs showed genome-wide significant association with risk to BA in the current cohort, while variants in other three loci show no association (*p* > 0.05). Additionally, the *ADD3* SNPs had smaller *p*-values and larger ORs compared with our previous results, which further supported the association of the *ADD3* locus with BA risk. We admit there are limitations of the study as a GWAS using a SNP chip; rare mutations in *ADD3* were not studied. However, if there were exome mutations causing BA, they would have been identified within the previous whole exome sequencing studies [1,35,36,37].

*ADD3* encodes γ-Adducin, which is an important regulator of both the spectrin-based membrane skeleton and the actin cytoskeleton [38]. Depletion of γ-Adducin attenuated cell–cell adhesions in model human epithelia [35]. It is reasonable to speculate that dysfunction of ADD3 might affect the membrane skeleton, cytoskeleton and cell–cell junction. Cholangiocytes with loosely formed cell–cell junctions were susceptible to damage from bile acid toxicity and viruses, which were suggested to be the underlying cause of BA [39]. We observed aberrant cytoplasmic deposition of ADD3. Future studies are needed to investigate the consequences of ADD3 dysfunction in cholangiocytes and hepatocytes. We did not observe auto-antibodies to ADD3 in the plasma of BA patients, which indicated that autoimmune response did not play a role in the contribution of ADD3 to the pathogenesis of BA. However, it should be noticed that the levels of autoantibodies in plasma vary according to the age of the patient. Our results were obtained from this particular analysis in 20 BA patients with a small age range, we could not exclude the possibility that autoantibodies to ADD3 attributed to fibrosis and cirrhosis in the later stage of BA. This is another limitation of our study.

A previous study showed that the expression level of *ADD3* was increased in BA livers [27]. This increase could occur via a genetic improvement of *ADD3*, since the risk alleles of the susceptibility SNPs rs17095355 and rs3862007 were associated with the increased expression of *ADD3* in multiple tissues from our eQTL analysis. Additionally, it is also possible that the increased expression resulted from secondary changes in the cholangiocytes in BA, such as cholestasis and/or fibrosis. Notably, previous observations showed that the risk alleles of rs17095355 were associated with reduced *ADD3* expression levels in 36 BA liver samples, which contrasted with the eQTL effect directions in liver and other tissues from healthy individuals. This inconsistency might be attributed to the spatio-temporal specificity of the regulatory effects of the associated SNPs, which need further clarification.

The HLA genes have one of the strongest genetic associations with autoimmunity [6]. HLA variants showed a nominal association with BA risk in our sample, which indicated that autoimmunity played an insignificant role in the pathogenesis of BA. Of note, the nonsignificant association might be caused by the following reasons. First, a moderate effect size with a value of OR of about 1.4 could not achieve a genome-wide significance in a case-control association study with the limited size of BA cases in the current study. Second, because of the likely genetic heterogeneity even in nonsyndromic BA, HLA genes may confer stronger susceptibility in specific subtypes of patients. Association study in mixed patients could attenuate the statistical power. The HLA region is the most polymorphic region in the human genome and the frequency distribution of HLA alleles is diverse in different human ethnic groups. Therefore, future case-control studies with a relatively large sample size in genetically homogeneous populations are needed to evaluate the genetic contribution of HLA variants to BA risk in other populations. Aberrant expression of HLA class II antigens on the affected organ also indicated a disease as being autoimmune in nature [6]. We observed ectopic expression of HLA-DR and DQ in the intrahepatic bile duct epithelium of BA livers. However, high expression of HLA-DR was also found in biliary epithelium of patients with other cholestatic liver disorders [40]. Thus, it was supposed that the high expression of HLA-II antigens in the biliary epithelium of BA livers was probably induced by cholestasis or some other factors. Taken together, our findings do not strongly support the role of autoimmunity in BA pathogenesis.

Nevertheless, our results were consistent with previous reports that HLA-DR and HLA-A variants showed association with BA susceptibility [22,24]. HLA class I molecules present endogenous peptides to CD8^+^ T cells and HLA class II molecules present exogenous peptides to CD4^+^ helper T cells. Our results found associations of HLA-DQB1 residue Ala57 and HLA-B Leu103 with BA risk with a nominal significance. The peptide is bound within the peptide binding groove of the MHC class I heavy chain, which is made up of two α-helices and a series of antiparallel β-sheets, which form the walls and floor of the groove, respectively. HLA-B Leu103 is located on β-sheets below the peptide. HLA-DQB1 Asp57 was associated with BA risk in our study. Interestingly, it was the most protective allele for type 1 diabetes but was found to increase the risk of Graves’ disease [41,42].

Although the current association results indicated that the HLA genes did not play a substantial role in the pathogenesis of BA. HLA genes might contribute to BA etiology by the mechanism of maternal microchimerism [43]. The presence of increased maternal microchimerism was found in the livers of BA infants, which might be caused by the increase in HLA compatibility between mother and child [43]. A previous study showed that HLA-A compatibility was significantly more frequent in BA patient–mother pairs than control–mother pairs. Additionally, the frequency of the most prevalent HLA haplotypes in a given population was significantly correlated with the incidence of BA in the population. These findings emphasize the importance of HLA in the etiopathology of BA [44].

Nonsyndromic BA was thought to be caused by a combination of genetic predisposition and environmental insults including viral infection (e.g., cytomegalovirus) or toxins (e.g., biliatresone) [8,45]. BA infants with CMV infection had poorer outcomes with a reduced clearance of jaundice, native liver survival and increased mortality [8]. In the current results, we found that no genetic variants were associated with CMV infection in BA patients. These lines of evidence might suggest that BA infants with or without CMV infection had the same genetic structure; CMV infection might worsen the outcomes of BA. However, it should be noted that the relatively small sample size of BA cases reduces the power of the study and undermines the conclusion. Further replication studies in independent sample sets are needed to validate the findings.

In conclusion, our study confirmed the association of *ADD3* with BA susceptibility and provided functional evidence for the contribution of *ADD3* to BA pathogenesis. Fine-mapping analysis of HLA genes revealed that two amino acids showed suggestive association with BA in the Chinese population. The lack of a significant association with HLA variants strengthens the case that an autoimmune response does not play a critical role in the pathogenesis of BA. CMV IgM (+) BA is not a distinct genetic subgroup.

## 4. Materials and Methods

### 4.1. Samples

From March 2007 to January 2019, 336 BA infants treated with Kasai surgery from Xinhua Hospital affiliated to Shanghai Jiaotong University School of Medicine were recruited in this study [15]. All participants were biologically unrelated Chinese individuals. BA was diagnosed by clinical manifestations, laboratory tests, imaging examinations and confirmed by operative cholangiography. We collected the clinical data of BA patients, including age at surgery, gender, CMV IgM results and liver biochemical indicators (bile acids, alanine aminotransferase (ALT), aspartate aminotransferase (AST), alkaline phosphatase (ALP), gamma-glutamyl transpeptidase (GGT), total bilirubin (TB) and direct bilirubin (DB)). Demographic information of patients is summarized in Appendix A. CMV IgM was determined by using LIAISON CMV kit (DiaSorin, Saluggia, Italy) according to the manufacturer’s protocol. According to the CMV serology results, the 317 BA infants that passed GWAS data quality control were classified into CMV IgM (+) group (n = 87) and CMV IgM (−) group (n = 141). The remaining 89 infants had no CMV IgM results. Blood samples were drawn from all participants after obtaining informed consent from their parents. Healthy controls with no history of BA and congenital disorders were recruited from individuals who underwent an annual health examination in Xinhua Hospital affiliated to Shanghai Jiaotong University School of Medicine and Capital Institute of Pediatrics. The median age of the control group is 37 (range 24–55 years). Both cases and controls are unrelated individuals of Han Chinese from east China. Genomic DNA was extracted from peripheral blood leukocytes using QIAamp DNA Blood Mini Kit (Qiagen, Hilden, Germany).

### 4.2. Genotyping and Quality Control for GWAS Data

A total of 336 BA subjects (the male to female ratio of 1.20:1) and 8900 control subjects (the male to female ratio of 1.25:1) were genotyped using the Illumina Infinium Global Screening Array. Genotype clustering was conducted with Illumina Ge-nomeStudio 2.0 software. Variants were removed if they had <98% success rate, had <0.01 minor allele frequency (MAF) or failed the Hardy–Weinberg test (*p* < 1.0 × 10^−6^) (Appendix A). Population structures of samples were evaluated via Principal Component Analysis (PCA) and Multidimensional Scaling (MDS) using SmartPCA and PLINK (Appendix A) [46,47]. Samples were excluded if they had a poor genotyping success rate (genotyping completeness <98%), showed an excess heterozygosity (being >3 standard deviations (s.d.) from the mean) or had outlying genetic ancestry (being >6 s.d. from the mean on PCA). PCA and MDS confirmed that all subjects who passed quality control were of Chinese ancestry and indicated a moderate degree of genetic stratification existed between cases and controls (λ_GC_ = 1.08) (Appendix A). Adjusting the top five principal components (PCs) as covariates via logistic regression and removing the associated SNPs in *ADD3* region reduced the extent of genome-wide inflation to 1.01, a level considered acceptable by conventional GWAS standards. PLINK [46] and R statistics packages were used for the calculation and visual plotting.

### 4.3. Imputation

Genotype imputation for *ADD3* gene region was performed using IMPUTE2 (v2.3.1) with the haplotypes derived from the 1000 Genomes Project Phase 1 (release v3) as reference data [48]. The imputed variants with imputation INFO score ≥ 0.9, MAF ≥ 0.01, call rate ≥ 98% and Hardy–Weinberg equilibrium (HWE) test with *p* ≥ 1 × 10^−6^ in the controls were saved for further analysis. Post-imputation association analysis was carried out using SNPTEST (v2.4.1) with the “expected” method adjusting for the first five PCs [49].

Imputation of two- and four-digit classical HLA alleles and amino acid polymorphisms of the HLA genes along with the SNPs that were not genotyped in the GWAS was performed using SNP2HLA (https://www.broadinstitute.org/mpg/snp2hla/ (accessed on 1 May 2017)) with the Pan-Asian data as the reference panel [30,31,32]. We applied post imputation QC criteria of minor allele frequency of ≥1% and PLINK INFO of ≥0.8 for the association analysis.

### 4.4. In Silico Functional Annotation of Associated Loci

The expression quantitative trait locus (eQTL) evidence of disease-associated SNPs was investigated using GTEx database Release V8 (dbGap Accession phs000424.v8.p2) [50]. To quantify the probability that the BA association and eQTL signals at a given locus were attributable to a shared causal variant, we conducted colocalization analysis by coloc using our association summary statistics and eQTLs data from GTEx [50]. eQTLs were extracted from GTEx database including liver (n = 208) and immune-related tissues, whole blood (n = 670) and spleen (n = 227). We restricted analyses to genes within 1 Mb of the associated SNPs of interest (eQTL *p* < 0.01 and m-value (Posterior Probability from METASOFT) > 0.9) and ran coloc.abf with default priors. We considered coloc tests with a Posterior Probability of hypothesis 4 (PPH4) ≥ 0.9 or PPH4.ABF > 0.5 as having strong or moderate evidence for colocalization.

### 4.5. Single-Cell RNA-Sequencing Data Sets and Analysis

In order to investigate the *ADD3* function in bile duct and liver development, we used an integration of single-cell RNA-sequencing (sc-RNAseq) datasets to explore the cell type specificity of *ADD3* expression [29]. The gene expression data across a variety of annotated parenchymal and non-parenchymal cells derived from 28 human healthy livers could be openly accessed from the datasets [29].

### 4.6. IF Analysis

We employed the IF method to determine the expression of *ADD3* in human liver tissues. Human liver tissue sections were obtained from BA infants and choledochal cyst (CC) infants who underwent surgery operation in Xinhua hospital. The BA liver tissues used for IF analysis were all obtained from patients with cholestasis and liver fibrosis. Human liver tissue specimens were fixed in 4% PFA. Rat anti-CK7 antibody (Developmental Studies Hybridoma Bank, Iowa City, IA, USA) diluted at 1:10, mouse anti-ADD3 antibody (Santa Cruz Biotechnology, Santa Cruz, CA, USA) diluted at 1:50 and DAPI (Beyotime Biotechnology, Shanghai, China) were used. Images were acquired by a fluorescence microscope (ECLIPSE Ni-E, Nikon, Chiyoda, Japan) and a Leica TCS SP8 confocal microscope (Leica Microsystems, Mannheim, Germany). The colocalization of ADD3 and CK7 in the images captured by a fluorescence microscope was determined and quantified using plugin ScatterJ in ImageJ 1.54f software. The qualifications were repeated three times, and representative scatterplots are displayed. Pearson’s correlation coefficient R value from 0.5 to 1 indicates positive co-localization, and an R value less than 0.5 indicates little co-localization.

### 4.7. Testing for the Autoantibody against ADD3

Enzyme linked immunosorbent assay (ELISA) was employed to measure the levels of auto-antibodies to ADD3 antigen in the plasma of BA patients and control patients. A total of 20 BA patients with (age: 1.7 ± 0.2 months, 60% male), 20 CC patients (age: 2.2 ± 0.3 months, 60% male) and 20 control patients (age: 72.7 ± 10.1 months, 60% male) without hepatobiliary and inflammatory diseases were recruited. The control patients were affected with umbilical hernia, inguinal hernia, hemangioma, constipatation or skin fistula. Blood samples were collected from these patients at the time of surgery. All the patients had signed informed consents.

The ADD3 antigen, a recombinant protein with the N-terminal His Tag (RPF203Hu01, Cloud-Clone Corp, Wuhan, China), was diluted at a concentration of 2 μg/mL in 50 μL of coating buffer (pH 9.6). A ninety-six-well plate (Labselect, Hefei, China) was coated with 100 ng of ADD3 antigen/well at 4 °C overnight. The wells were washed three times with PBST (PBS with 0.1% Tween 20) and blocked with 2% BSA in PBST for 1 h at room temperature. After three times of wash, human plasma samples diluted at 1:500 in PBS were added in the wells and incubated for 2 h at room temperature. After washing five times with PBST, 100 μL of a 1:3000 dilution of horseradish peroxidase (HRP)-conjugated goat anti-human IgG (H + L) (SA00001-17, Proteintech, Rosemont, IL, USA) antibody was added to each well and incubated for 1 h at room temperature, and the plate was then washed 4 times. The TMB peroxidase Substrate (100 μL/well) was added and incubated for 15 min at 37 °C and terminated the color reaction with stop solution (50 μL/well). The optical density reading was measured immediately at 450 nm using Synergy Hybrid Reader (BioTek Instruments, Winooski, VT USA). All tests were performed in triplicate. Data were analyzed using independent samples *t*-test.

## Figures and Tables

**Figure 1 ijms-24-14719-f001:**
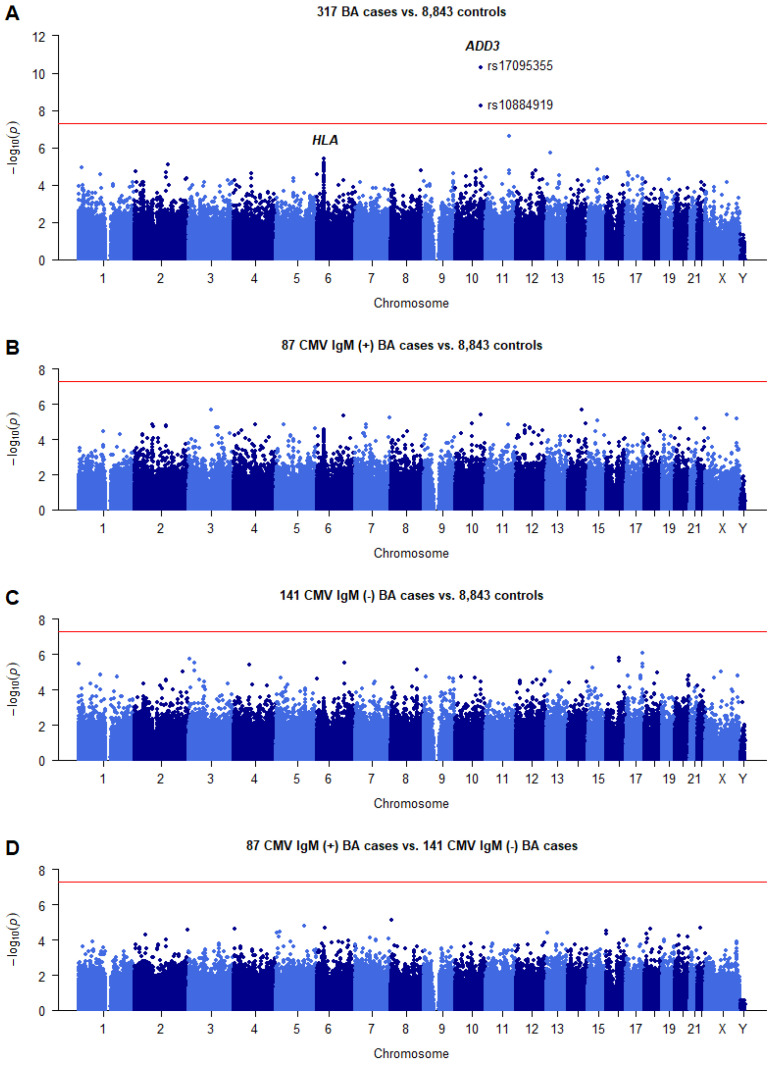
Summary of results from the genome-wide association study for BA risk in Chinese. (**A**) Association results of 317 BA cases compared to 8843 controls. (**B**) Association results of 87 CMV IgM+ BA cases compared to 8843 controls. (**C**) Association results of 141 CMV IgM− BA cases compared to 8843 controls. (**D**) Association results of 87 CMV IgM+ BA cases compared to 141 CMV IgM− BA cases. Dots represent the −log_10_
*p* values of the Cochran–Armitage trend test from 398,532 SNPs. The genome-wide significance threshold (*p* = 5 × 10^−8^) is indicated by the red horizontal line.

**Figure 2 ijms-24-14719-f002:**
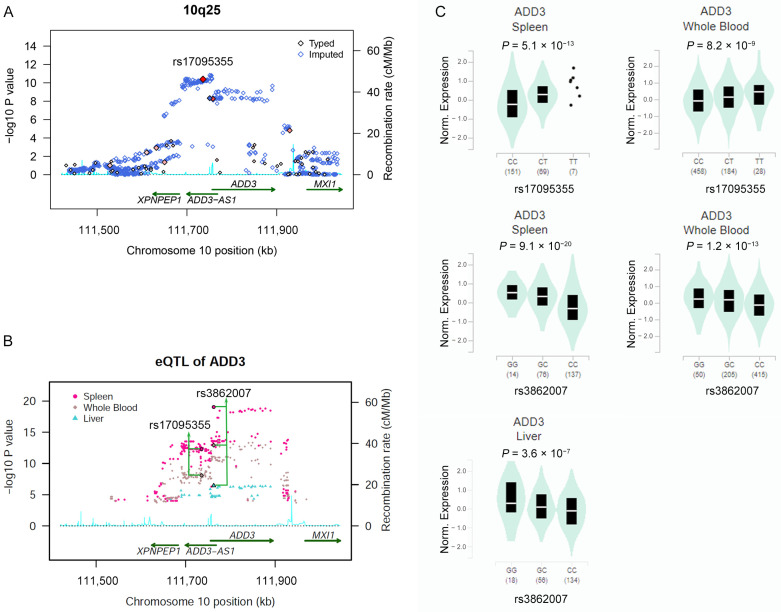
Plots of the association results in the *ADD3* region at 10q24 and the eQTLs of *ADD3*. (**A**) Association results for both genotyped and imputed SNPs in the *ADD3* region at 10q24. The diamonds with black border represent the genotyped SNPs and the diamonds with blue border represent the imputed SNPs. The color intensity (write to red) of each genotyped SNP spot reflects its r^2^ with the top SNP rs17095355. Genetic recombination rates, estimated using the HapMap CHB samples, are shown in cyan. (**B**) The eQTLs data were obtained from genotype and transcriptome data of the GTEx Analysis Release V8. The eQTLs are genotyped or imputed SNPs from GWAS data. (**C**) The risk alleles of rs3862007 and rs17095355 are both associated with increased levels of *ADD3* expression.

**Figure 3 ijms-24-14719-f003:**
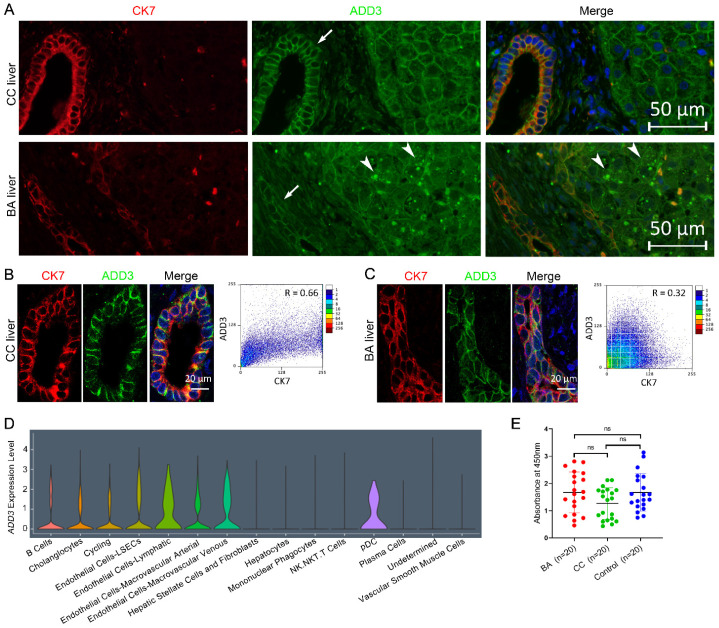
Expression pattern of ADD3 and assays of anti-ADD3 auto-antibodies. (**A**) IF detection of ADD3 in combination with CK7 in choledochal cyst (CC) and BA liver sections. The perimembranous staining of ADD3 in CC cholangiocytes is more intense than that in BA (arrows). ADD3 deposits are observed in cytoplasm of hepatocytes in BA livers (12/12) (arrow heads). Red color: the cholangiocyte marker CK7; green color: staining for ADD3; blue color: DAPI staining of nuclei. 400× magnification. (**B**,**C**) Colocalization analysis of ADD3 and CK7 in cholangiocytes of CC and BA infants. ADD3 and CK7 were colocalized at the same spots beneath the cell membrane of CC cholangiocytes. Pearson’s correlation coefficient (R) between the green foci and red foci is shown in scatter diagram (right); 1000× magnification. (**D**) The expression profiles of *ADD3* from ScRNAseq data. (**E**) ELISA assays to detect anti-ADD3 auto-antibodies in plasma of BA patients. The plasma samples from 20 control patients without inflammation (blue), 20 BA patients (red) and 20 choledochal cyst patients (CC, green) were measured for IgG binding to ADD3. The levels of anti-ADD3 auto-antibodies in plasma BA samples showed no differences between groups. ns: not significant.

**Figure 4 ijms-24-14719-f004:**
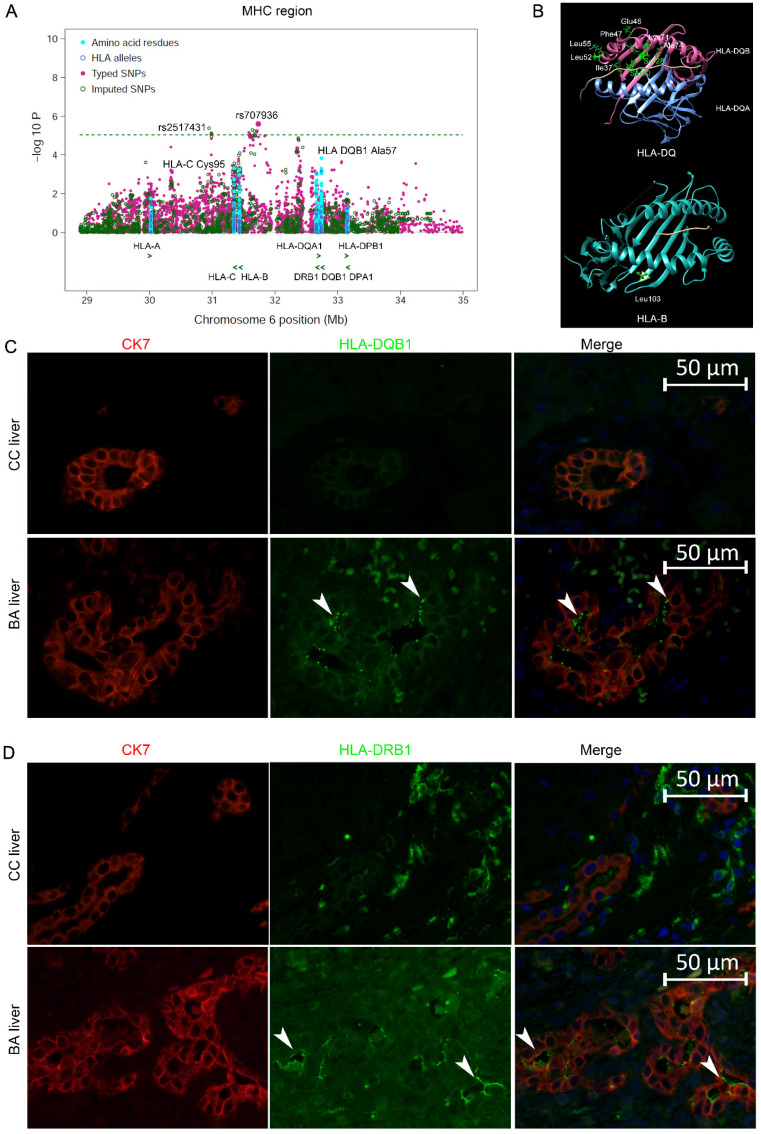
Regional plots of *HLA* loci association with BA risk and expression of HLA-DQB1 and DRB1 in BA livers. (**A**) Regional plots of *HLA* loci association with BA risk. Each dot represents the −log_10_
*P* of the variants, including typed SNPs, imputed SNPs, classical HLA alleles and HLA amino acid residues. The green horizontal dashed line represents *p* = 1 × 10^−5^. The top signal was from rs707936 (OR = 1.68, *P*_logistic_ = 3.57 × 10^−6^). The best associated residue was HLA-DQB1 Ala57. The green arrowheads indicate the position and the direction of transcription of the respective genes. (**B**) The best associated residues of HLA-B and HLA-DRQB1. The protein structures of HLA-DQ and HLA-B are based on Protein Data Bank (PDB) entries 7KEI and 5WMR, respectively, which were prepared using Chimera 1.11.2. (**C**,**D**) IF detection of HLA-DQB1 and DRB1 in combination with CK7 in CC and BA liver sections. Both HLA-DQB1 and DRB1 were detected on the cell membrane of cholangiocytes of BA livers (12/12) (arrow heads). Red color: the cholangiocyte marker CK7; green color: staining for ADD3; blue color: DAPI staining of nuclei; 400× magnification.

**Table 1 ijms-24-14719-t001:** Association results of SNPs in *ADD3* and *HLA* gene region with BA risk.

Gene	SNP	Genotype/Allele	Genotype/Allele Distribution N (%)		*p* Value	
			Case	Control	Logistic	Genotype
*ADD3*	rs17095355	TT	83 (26.2)	1328 (15.0)	4.83 × 10^−11^	3.16 × 10^−10^
		TC	161 (50.8)	4164 (47.1)		
		CC	73 (23.0)	3351 (37.9)		
		T	327 (51.6)	6820 (38.6)		
		C	307 (48.4)	10,866 (61.4)		
*ADD3*	rs10884919	TT	73 (23.1)	1212 (13.7)	5.74 × 10^−9^	3.54 × 10^−8^
		TC	160 (50.6)	4117 (46.6)		
		CC	83 (26.3)	3511 (39.7)		
		T	306 (48.4)	6541 (37.0)		
		C	326 (51.6)	11,139 (63.0)		
*HLA*	rs707936	AA	7 (2.2)	88 (1.0)	3.46 × 10^−6^	2.8 × 10^−5^
		AG	82 (25.9)	1517 (17.2)		
		GG	228 (71.9)	7232 (81.8)		
		A	96 (15.1)	1693 (9.6)		
		G	538 (84.9)	15,981 (90.4)		

**Table 2 ijms-24-14719-t002:** Associations of HLA amino acid residues and alleles with BA risk in Han Chinese.

Genes	Effective Residue/Allele	PLINK INFO	ERF (%)		OR (95% CI)	*p* Value
			Cases (n = 317)	Controls (n = 8843)		
HLA-DQB1	Ala57	0.97	0.24	0.18	1.44 (1.20–1.74)	1.23 × 10^−4^
HLA-DQB1	Asp57	0.99	0.62	0.69	0.75 (0.63–0.88)	4.32 × 10^−4^
HLA-DQB1	Val57	1.00	0.08	0.07	1.10 (0.83–1.47)	5.09 × 10^−1^
HLA-DQB1	Ser57	1.02	0.06	0.06	0.99 (0.71–1.38)	9.58 × 10^−1^
HLA-DQB1	Ala74	0.99	0.17	0.12	1.46 (1.18–1.81)	4.58 × 10^−4^
HLA-DQB1	Glu74	0.99	0.63	0.68	0.82 (0.70–0.97)	1.83 × 10^−2^
HLA-DQB1	Ser74	0.99	0.20	0.20	0.99 (0.81–1.21)	9.11 × 10^−1^
HLA-DQB1	Lys71	0.99	0.17	0.12	1.46 (1.18–1.81)	4.58 × 10^−4^
HLA-DQB1	Thr71	0.99	0.63	0.68	0.82 (0.70–0.97)	1.83 × 10^−2^
HLA-DQB1	Asp71	0.99	0.05	0.06	0.91 (0.64–1.31)	6.22 × 10^−1^
HLA-DQB1	Ala71	1.00	0.15	0.15	1.02 (0.82–1.28)	8.48 × 10^−1^
HLA-DQB1	Leu52	0.99	0.17	0.12	1.46 (1.18–1.81)	4.58 × 10^−4^
HLA-DQB1	Phe47	0.99	0.17	0.12	1.46 (1.18–1.81)	4.58 × 10^−4^
HLA-DQB1	Glu46	0.99	0.17	0.12	1.46 (1.18–1.81)	4.58 × 10^−4^
HLA-DQB1	Ile37	0.99	0.17	0.12	1.46 (1.18–1.81)	4.58 × 10^−4^
HLA-DQB1	Tyr37	0.95	0.75	0.80	0.77 (0.64–0.93)	5.18 × 10^−2^
HLA-DQB1	Asp37	0.90	0.08	0.08	0.99 (0.73–1.32)	9.25 × 10^−1^
HLA-DQB1	Ser30	0.99	0.17	0.12	1.46 (1.18–1.81)	4.58 × 10^−4^
HLA-DQB1	Tyr30	1.00	0.67	0.70	0.84 (0.71–0.99)	3.72 × 10^−2^
HLA-DQB1	His30	0.99	0.17	0.18	0.95 (0.76–1.17)	6.06 × 10^−1^
HLA-DQB1	Ser28	0.99	0.17	0.12	1.46 (1.18–1.81)	4.58 × 10^−4^
HLA-DQB1	0201	0.99	0.17	0.12	1.46 (1.18–1.81)	4.75 × 10^−4^
HLA-B	Leu103	0.98	0.34	0.28	1.34 (1.13–1.58)	6.76 × 10^−4^
HLA-B	5801	1.00	0.09	0.06	1.58 (1.19–2.10)	1.62 × 10^−3^
HLA-DRB1	Ala71	1.03	0.11	0.15	0.65 (0.50–0.84)	9.29 × 10^−4^
HLA-DRB1	Lys71	0.99	0.09	0.06	1.49 (1.12–1.98)	6.03 × 10^−3^
HLA-DRB1	Arg71	1.00	0.77	0.74	1.16 (0.96–1.39)	1.32 × 10^−1^
HLA-DRB1	Glu71	1.00	0.04	0.05	0.89 (0.60–1.32)	5.62 × 10^−1^
HLA-DRB1	15	1.02	0.11	0.15	0.65 (0.50–0.84)	9.41 × 10^−4^

HLA, human leucocyte antigen; OR, odds ratio; 95% CI, 95% confidence interval; ERF, effective residue frequency; CI, confidence interval; OR, odds ratio.

## Data Availability

Data that are presented in this study are available upon request from the corresponding author.

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
