# Peer review of "Contribution of ADD3 and the HLA Genes to Biliary Atresia Risk in Chinese"

_ijms, 2023, doi:10.3390/ijms241914719_

Round 1
Reviewer 1 Report
Dear authors, I recently had the opportunity to review your article titled " Contribution of ADD3 and the HLA genes to biliary atresia risk in Chinese" First and foremost, I find these findings contribute to our understanding of the genetic factors and pathogenesis of BA susceptibility, particularly in the context of Chinese infants.
While the manuscript contains valuable insights and potential contributions to the field, there are several areas that require substantial improvement to meet the standards of publication. I believe that addressing these issues will significantly enhance the quality and impact of the research.
Therefore, I consider this manuscript not suitable for publication in IJMS. Nonetheless, I suggest to authors some feedback suggestions apply to other journals.
- In the abstract, it is advisable to provide more context for the study and to outline the methods employed in the research. Furthermore, the conclusions in the abstract should be more explicitly linked to the study's findings.
- The figure 2C is not clearly visible as it appears quite small, making it difficult to interpret the information presented in the manuscript.
- In Figure 2D, the color blue is not mentioned within the figure, and the image lacks scales for reference. It's essential to include scales in both the images and the figure legend for clarity. Additionally, I recommend including a negative fluorescence control, as the fluorescence levels of ADD3 appear quite low, and there may be potential interference from tissue autofluorescence.
- Regarding Figure 2F, it is somewhat challenging to understand. It would be helpful to include a more illustrative figure that better represents the ELISA results for improved clarity.
- In Figure 2D, it appears that the arrowheads are missing. Including arrowheads would help to pinpoint and highlight specific features or areas of interest in the image, enhancing the clarity of the figure.
- n Figure 3, the legend is missing a description for Figure 3D, and the scales for the images are not provided. Including this information in the legend would be beneficial for a more comprehensive understanding of the figure.
- It is recommended to include images from confocal microscopy to visualize colocalization of the markers. Additionally, performing a quantification analysis of colocalization would provide valuable insights into the results.
Reviewer 2 Report
This latest manuscript appears to build on the results of previous work published by the same research group (which is only slightly apparent when perusing the manuscript). This needs to be made clearer both in the Introduction and Discussion sections.
Plus, it seems that different controls were used for different aspects of this study. Can the authors please provide an explanation as to how the 8900 control subjects for the genotyping and quality control for GWAS data were sourced? Plus what were the demographics of the controls, for eg median age (and range). Were they all completely healthy?
For the liver tissue sections that were obtained for immunofluorescence analysis, were these all obtained at a similar time point during the patients course with biliary atresia or were some of these patients presenting a lot later? The reason this question is asked, is that again the results that are obtained for the immunofluorescence may be influenced by what stage the patient presents (ie with severe cholestasis alone, with liver fibrosis, with cirrhosis).
Is there any chance that the differences in the ages of the index patients (with biliary atresia) versus the control patients in the subgroup where ELISA was used to measure the levels of autoantibodies to ADD3 in plasma, contributed to the results that were seen. The results that could be obtained for this particular analysis could potentially vary according to the age of the patient. Is this not another limitation of this study?
Round 2
Reviewer 1 Report
In my opinion the manuscript has improved enough to be published. I thank the authors for all their modifications in order to solve the previous deficiencies.
Reviewer 2 Report
The authors have responded appropriately to the reviewers comments and have amended the manuscript accordingly